# SARS-CoV-2 pneumonia follow-up and long COVID in primary care: A retrospective observational study in Madrid city

**Sara Ares-Blanco**[1], **Marta Pérez Álvarez**[1], **Ileana Gefaell Larrondo**[1], **Cristina Muñoz**[2], **Vanesa Aguilar Ruiz**[1], **Marta Castelo Jurado**[1], **Marina Guisado-Clavero**[3]*

**1** Federica Montseny Primary Health Care Center, Gerencia Asistencial Atención Primaria, Madrid Health Service, Madrid, Spain, **2** Servicio de Atención Rural Norte, Gerencia Asistencial Atención Primaria, Madrid Health Service, Madrid, Spain, **3** Unidad Docente Multiprofesional de Atención Familiar y Comunitaria Norte, Gerencia Asistencial Atención Primaria, Madrid Health Service, Madrid, Spain

* marina.guisado@salud.madrid.org

**Data Availability Statement:** Data cannot be made freely available in the manuscript, the supplemental

## Abstract

### Background

Patients with COVID-19 are follow-up in primary care and long COVID is scarcely defined. The study aim was to describe SARS-CoV-2 pneumonia and cut-offs for defining long COVID in primary care follow-up patients.

### Methods

A retrospective observational study in primary care in Madrid, Spain, was conducted. Data was collected during 6 months (April to September) in 2020, during COVID-19 first wave, from patients $\geq$ 18 years with SARS-CoV-2 pneumonia diagnosed. Variables: sociodemographic, comorbidities, COVID-19 symptoms and complications, laboratory test and chest X-ray. Descriptive statistics were used, mean (standard deviation (SD)) and medians (interquartile range (IQR)) respectively. Differences were detected applying $X^2$ test, Student's *T*-test, ANOVA, Wilcoxon-Mann-Whitney or Kruskal-Wallis depending on variable characteristics.

### Results

155 patients presented pneumonia in day 7.8 from the onset (79.4% were hospitalized, median length of 7.0 days (IQR: 3.0, 13.0)). After discharge, the follow-up lasted 54.0 median days (IQR 42.0, 88.0) and 12.2 mean (SD 6.4) phone calls were registered per patient. The main symptoms and their duration were: cough (41.9%, 12 days), dyspnoea (31.0%, 15 days), asthenia (26.5%, 21 days). Different cut-off points were applied for long COVID and week 4 was considered the best milestone (28.3% of the sample still had symptoms after week 4) versus week 12 (8.3%). Patients who still had symptoms >4 weeks follow-up took place over 81.0 days (IQR: 50.5, 103.0), their symptoms were more prevalent and lasted longer than those $\leq$ 4 weeks: cough (63.6% 30 days), dyspnoea (54.6%, 46 days), and asthenia (56.8%, 29 days). Embolism was more frequent in patients who still had symptoms >4 weeks than those $\leq$4 weeks (9.1% vs 1.8%, *p* value 0.034).

files, or a public repository due to ethical restrictions. Researchers who wish to gain access to the data can write to the corresponding author. The study protocol was inspected and approved by the Ethics Committee of University Hospital Gregorio Marañón (13/2020 and 24/2020), the Primary Care Southeast Research Committee (CLISE) at Madrid region (09/20) and the Spanish Drug Agency (AEMPS). Requests for access to the data may be sent to the ethical committee (ceim.hgugm@salud.madrid.org) and the research committee (udsureste@salud.madrid.org), which is subject to prior determination of the terms and conditions of the request and in compliance with the applicable regulations.

**Funding:** This manuscript has received a grant from the Foundation for Biomedical Research and Innovation in Primary Care (FIIBAP) for publication in 2021 (FIIBAP/mar21).

**Competing interests:** The authors declare that they have no competing interests.

## Conclusion

Most patients with SARS-CoV-2 pneumonia recovered during the first 4 weeks from the beginning of the infection. The cut-off point to define long COVID, as persisting symptoms, should be between 4 to 12 weeks from the onset of the symptoms.

## Introduction

The COVID-19 pandemic, caused by SARS-CoV-2 virus, has affected more than 119 millions of patients worldwide until 14th March 2021 [1]. At the beginning of the pandemic, research was focused on the virus description, the epidemiology, the clinical syndrome and possible treatments. The evolution of the disease was described as a three weeks course until discharge in severe cases [2]. The initial approach was to describe the disease until discharge [3] but some patients still had symptoms after the acute phase, as fatigue, dyspnoea, cough and joint pain [4, 5], including oxygen therapy at home after discharge [6]. This condition has been described as long COVID [7] but patients' perspective have not been enough considered [8].

Public Health England defined long COVID as continuous symptoms for ≥8 weeks after discharge [9] but Greenhalgh *et al.* described it as ≥ 3 weeks after the onset of COVID-19 symptoms [10]. NICE describes long COVID as symptoms that continue after the acute phase, they distinguished ongoing symptomatic COVID-19 (from 4 to 12 weeks) and post-COVID-19 (≥12 weeks) [11]. This approach was criticized by Gorna *et al.* who prefer the 4 week milestone to define it [12]. Cut-off point to define long COVID is controversial but patient's perspective should be considered. Besides, most of the COVID-19 data were obtained from hospitalized patients and there are few studies from outpatient clinics [13]. How to address persistent complications is still unknown, but primary care can follow-up them with support of specialist assessment if necessary [10, 14].

In Spain, there have been 3,183,704 COVID-19 cases until the 14th of March 2021 [15]. One of the epicentres of the disease was in the Madrid region, where the health system collapsed during the first month of the pandemic [16]. The national seroprevalence study showed that 11.3% of Madrid residents had antibodies against SARS-CoV-2 in the first wave [17]. In the region, mild COVID-19 cases were managed at home; only patients with SARS-CoV-2 pneumonia or seriously illness were referred to hospital. The data in Madrid showed that 99,723 patients were admitted to hospitals but 730,052 cases were managed by home or remote assessment by primary care teams by 14th of March 2021 [16]. The primary care follow-up recommended close remote assessment for patients by their GP or nurse to until their recovery. The research aimed to determine patient and disease characteristics associated with SARS-CoV-2 pneumonia since the pneumonia diagnosis until recovery in a study population aged 18 to 91 years during the follow-up in primary care especially focusing on patients who still had symptoms 4 weeks after SARS-CoV-2 pneumonia diagnosis. The secondary outcome was to describe the different cut-off points of long COVID definition in week 4 and in week 12.

## Materials and methods

### Study design and setting

A retrospective observational study was conducted in a primary health care centre (PHCC) named Federica Montseny PHCC located in Madrid (Spain). Medical records were

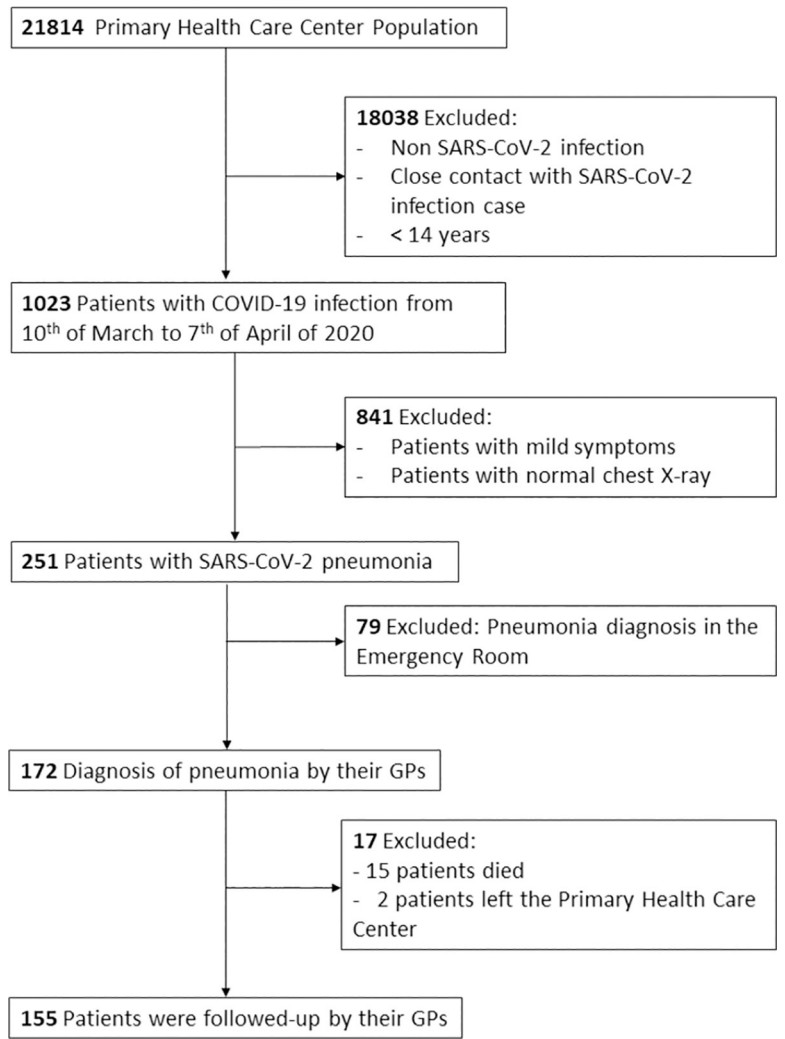

**Fig 1. Flow chart.**

obtained from the electronic health records of 14 GPs, who provide medical care to 19,061 patients.

## Selection of study subjects

The PHCC provided care to 21814 population, from whom 1,023 patients were diagnose from COVID-19 (confirmed by RCP or symptoms compatible with COVID-19), from the 10th of March to the 7th of April of 2020. Pneumonia was present in 251 and 155 were diagnosed and followed-up by Federica Montseny PHCC during 6 months (Fig 1).

Throughout the first wave, GPs evaluated SARS-CoV-2 clinical course by a narrow follow-up. Those with suspicious of SARS-CoV-2 pneumonia (persisting cough and/or fever and/or dyspnoea and/or chest pain) where remitted to the radiology service for a chest-X-ray. Urgent chest-X-ray is routinely available in our PHCC, radiologists evaluate every exam in the next 2 hours to give a proper report about the findings. Patients with lung affection by SARS-CoV-2 infection, were diagnosed of pneumonia by SARS-CoV-2 and were recruited consecutively for the study. Data was collected 6 months after the diagnosis, until 30th of September. Eligibility

criteria were considered in patients $\geq$ 18 years, followed-up by the primary care practice during the next 6 months after SARS-CoV-2 pneumonia diagnosis, with a suspicious of SARS--CoV-2 pneumonia and a radiologist report which confirmed the SARS-CoV-2 pneumonia. Exclusion criteria were applied to those patients with SARS-Co-V-2 pneumonia who were not followed-up by the Madrid Public Health System (for instance, they could move on to another region or they used private insurance) or those who did not survive during hospital length.

## Measurements

Data were collected by self-reporting information from GPs clinical courses. To decide which medical records and SARS-CoV-2 related outcomes were needed (physical symptoms, complementary examinations), we observed clinical guideline from WHO and Spanish Ministry of Health [1, 15]. Then, variables collected were: sociodemographic characteristics (sex, age, ethnicity), comorbidities (hypertension, dyslipidaemia, body mass index (BMI) $\geq$ 25 Kg/m$^2$, diabetes, asthma, chronic obstructive pulmonary disease (COPD), chronic kidney disease, heart failure, cancer), COVID-19 symptoms (temperature $\geq$ 37.5˚C, dyspnoea, cough, gastrointestinal disturbances, chest pain, asthenia, myalgia, headache, odynophagia, rhinitis, anosmia, dysgeusia), pneumonia radiological pattern (unilateral or bilateral) or normal radiology after the acute phase, laboratory tests (C-reactive protein, lymphocytes, ferritin, D-dimer, detection of viral RNA via reverse transcription-polymerase chain reaction -RT-PCR-), need for oxygen after diagnosis and during the follow-up, days of evolution of the disease (SARS-CoV-2 pneumonia diagnosis from the onset of the symptoms, hospital admission, follow-up in primary care), and number of calls from primary care until recovery. We defined the course of COVID-19 if resolution of the symptoms were within 4 weeks since the first symptom appeared and if the persistence of symptoms after 4 weeks since the SARS-CoV-2 pneumonia diagnosis (12). Data collection and study design has been performed to avoid possible bias.

## Statistical analysis

In descriptive statistics, continuous variables were assessed as means with standard deviation (SD) or by medians with interquartile range (IQR), categorical variables were assessed as percentages. Statistical association in baseline characteristics were calculated using $X^2$ test for categorical variable, Student's $T$-test or ANOVA for normally distributed variables and Wilcoxon-Mann-Whitney or Kruskal-Wallis test for non-Normal variables. Besides, data were stratified by age group (<50 years, 50–75 years, $\geq$75 years). The association between the 12 weeks cut-off point versus 4 weeks cut-off point was assessed using logistic regression to calculate OR with 95% confidence interval (CI). Bias control has been performed during study designs and data collection. All analyses were conducted using STATA (version 16). All data were anonymised.

## Ethical approval

This study was approved by the Ethics Committee of University Hospital Gregorio Marañón (13/2020 and 24/2020), the Primary Care Southeast Research Committee (CLISE) at Madrid region (09/20) and the Spanish Drug Agency (AEMPS). The informed consent was not required because of the anonymization of data and retrospective collected data.

## Results

Of the 177 SARS-CoV-2 pneumonia cases that were diagnosed by GPs during the study period, 155 were included since they were followed-up until recovery at the practice; the mean

**Table 1. SARS-CoV-2 pneumonia patient's characteristics (N = 155) followed by primary care in 2020 from onset to recovery (6 months).**

| Patient´s characteristics | | Symptoms | | | Laboratory tests and chest X-ray | | | Admission and Follow-up | |
|---|---|---|---|---|---|---|---|---|---|
| **Sociodemographic** | | **Symptoms** | **Until pneumonia diagnosis** | **During follow-up** | **Blood test** | **At pneumonia diagnosis** | **During follow-up** | **Hospital admission** | |
| Age (years)# | 58.8 (16.7) | Fever (>37,5°C)& | 131 (84.5) | 11 (7.1) | mean days# | 8.0 (4.1) | 35.2 (14.9) | Patients who were admitted& | 123 (79.4) |
| Sex (women)& | 80 (51.6) | Cough& | 130 (83.9) | 65 (41.9) | Lymphocytes (10E3/µL)$ | 1200.0 (850.0, 1500.0) | 2434.6 (916.8) | Length stay$ | 7.0 (3.0, 13.0) |
| Foreigner& | 36 (23.2) | Dyspnoea& | 91 (58.7) | 48 (31.0) | D-dimer (µg/L)$ | 424.5 (287.0, 910.0) | 460.0 (260.0, 850.0) | ICU entry& | 4 (2.6) |
| **Comorbidies** | | Myalgia& | 47 (30.3) | 22 (14.3) | Fibrinogen >500 (mg/dL)& | 114 (73.5) | 12 (11.4) | **Complications** | |
| Overweight (BMI >25)& | 84 (53.8) | Asthenia& | 40 (25.8) | 41 (26.5) | Ferritin (µg/L)$ | 439.0 (200.0, 1215.0) | 153.0 (50.0, 401.0) | Thromboembolism& | 6 (3.9) |
| Hypertension& | 71 (45.8) | Headache& | 40 (25.8) | 18 (11.6) | CRP (mg/L)$ | 60.01(22.6, 124.4) | 1.2 (0.3, 3.4) | Readmission& | 9 (5.8) |
| Dyslipidaemia& | 60 (38.7) | Chest pain& | 25 (16.1) | 25 (16.1) | **Chest X-ray** | **Type of pneumonia diagnosis** | **Normal Chest-Xray during follow-up** | Home Oxygen Therapy& | 8 (5.2) |
| Type II Diabetes& | 29 (18.7) | Dysgeusia& | 8 (5.2) | 3 (1.9) | mean days# | 8.0 (4.1) | 53.0 (29.3) | **Follow-up at Primary Care** | |
| Asthma& | 20 (12.9) | Anosmia& | 2 (1.3) | 4 (2.6) | Unilateral Pneumonia& | 45 (29.0) | 32 (71.1) | Follow-up (days)$ | 54.0 (42.0, 88.0) |
| Smoke habit& | 12 (8.3) | GI symptoms& | 67 (43.2) | 41 (26.5) | Bilateral pneumonia& | 110 (71.0) | 64 (58.1) | Number of phone calls# | 12.2 (6.4) |
| Chronic Kidney Disease& | 11 (7.1) | **Number of symptoms** | | | | | | | |
| Cancer& | 10 (6.5) | None& | 0(0) | 27 (17.4) | | | | | |
| COPD& | 8 (5.2) | 1–3& | 83 (53.9) | 87 (56.1) | | | | | |
| Heart failure& | 6 (3.9) | ≥ 4& | 71 (46.1) | 41 (26.5) | | | | | |

&: number (%)

#: mean (standard deviation)

$:median (interquartile range), BMI (body mass index), COPD (chronic obstructive pulmonary disease), GI (gastrointestinal), ICU (intensive unit care), CRP (C reactive protein).

age was 58.8 years (SD 16.7), 51.6% were female. Table 1 provides an overview of the baseline characteristics of the whole sample (stratified by age subgroups can be found in S1 File). Regarding admissions, 79.5% were admitted and 20.5% were treated on an outpatient basis. The median hospital length of stay was 7.0 days median (IQR: 3.0, 13.0) and 4 patients (2.6%) were admitted to intensive care. Once they were discharged from the hospital, they were followed-up in primary care with a median of 54.0 days (IQR: 42.0, 88.0) and 5.8% of the patients required hospital re-admission because of COVID-19 complications. A mean of 12.2 (SD: 6.4) calls were registered by their GP and nurse to monitor them. The evolution of the disease is described in Fig 1.

## Follow-up in primary care

In the follow-up, cough was the most frequent symptom (41.9%), followed by dyspnoea (31.0%), gastrointestinal disturbances and asthenia (26.5%). Cough, dyspnoea and asthenia were the most prevalence symptoms after hospital discharge (41.9%, 30.0% and 26.5%

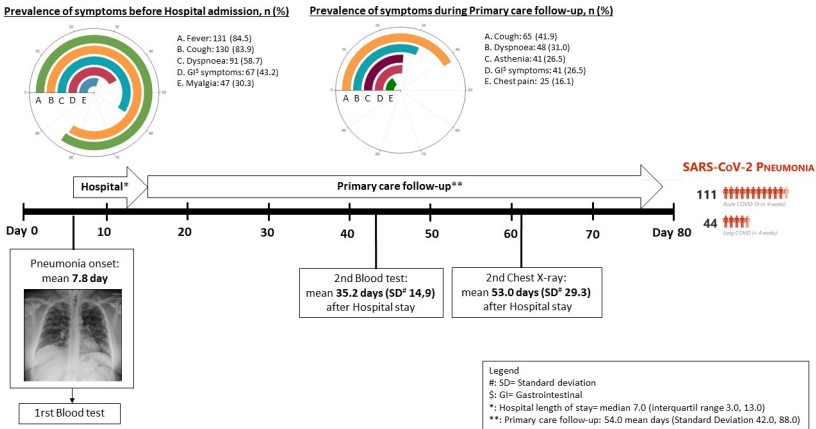

**Fig 2. Clinical course of SARS-CoV-2 pneumonia 6 months follow-up in primary care (2020).**

respectively) (Fig 2). Patients who were asymptomatic were a minority (17.4%); most of the patients had among 1 to 3 symptoms (56.1%).

Routine laboratory testing was performed at 35.2 day of media (SD: 14.9) after the SARS-CoV-2 pneumonia diagnosis. All acute phase reactants improved except for D-dimer, which increased lightly from 424.5 µg/L at the pneumonia diagnoses to 460.0 µg/L in the routine test (Table 1). Patients often had normal lymphocytes (1,200.0 x10E3/ÂµL) during the acute illness but in the routine test, their level increased to 2434.6x10E3/ÂµL. Of the 155 enrolled patients, 147 patients (94.3%) underwent a routine follow-up chest X-ray. The chest X-ray was performed on mean day 53.0 (SD: 29.3) after the SARS-CoV-2 pneumonia diagnosis, 64.1% had a normal chest X-ray.

## Long COVID definitions and characteristics

As the definition of long COVID remains uncertain, we explored the prevalence of long COVID in our population with different cut-off points from week 3 to week 12. The prevalence was 34.1% in week 3 but 8.3% in week 12 (Table 2). We decided to take the 4 weeks milestone as the definition of long COVID for this article because time-limit allowed a practical approach to patients. Table 3 shows characteristics of patients as they were classified as ≤ 4

**Table 2. Possible cut-off points to define long COVID.**

| Length long COVID (weeks) | Patients (n, %) | CI 95% |
|---|---|---|
| 3 | 53 (34.1) | 0.26–0.41 |
| 4 | 44 (28.3) | 0.21–0.35 |
| 5 | 33 (21.2) | 0.14–0.27 |
| 6 | 26 (16.7) | 0.10–0.22 |
| 7 | 24 (15.4) | 0.09–0.21 |
| 8 | 20(12.9) | 0.09–0.18 |
| 9 | 18 (11.6) | 0.06–0.17 |
| 10 | 17(10.9) | 0.05–0.15 |
| 11 | 16(10.3) | 0.05–0.15 |
| 12 | 13(8.3) | 0.03–0.12 |

n (number), CI (confidence interval).

**Table 3. Differences between patients with SARS-CoV-2 pneumonia clinical course of ≤4 weeks and >4 weeks (long COVID) from a 6 months follow-up in primary care (2020).**

| Patient´s characteristics | | | | Follow-up Symptoms | | | | Follow-up in primary care | | | |
|---|---|---|---|---|---|---|---|---|---|---|---|
| Sociodemographic | ≤4 weeks | >4 weeks | p-value | Symptoms | ≤4 weeks | >4 weeks | p-value | Primary Care | ≤4 weeks | >4 weeks | p-value |
| n | 111 | 44 | | Fever (>37,5°C)[&] | 6 (5.4) | 5 (11.1) | 0.210 | Days of follow-up[$] | 50.0 (39.0, 73.0) | 81.0 (50.5, 103.0) | <0.001 |
| Sex (female)[&] | 52 (46.8) | 28 (63.6) | 0.050 | Cough[&] | 37 (33.3) | 28 (63.6) | <0.001 | Number of phone calls by GP or nurse during the follow-up[#] | 10.6 (5.0) | 16.0 (8.0) | <0.001 |
| Age (mean)[&] | 60.1 (17.6) | 55.7 (14.2) | 0.140 | Dyspnoea[&] | 24 (21.6) | 25 (54.6) | <0.001 | **Blood Test, day** | 34.9 (25.5) | 36.1 (13.6) | |
| **Comorbidities** | | | | Myalgia[&] | 10 (9.1) | 12 (27.3) | 0.004 | Lymphocytes (10E3/µL)[$] | 2492.6 (956.8) | 2277.1 (789.7) | 0.240 |
| Overweight (BMI >25)[&] | 61 (55.0) | 22 (50.0) | 0.370 | Asthenia[&] | 16 (14.4) | 25 (56.8) | <0.001 | D-dimer (µg/L)[$] | 460.0 (260.0, 870.0) | 465.0 (280.0, 730.0) | 0.730 |
| Hypertension[&] | 59 (53.2) | 12 (27.3) | 0.004 | Headache[&] | 7 (6.3) | 11 (25.0) | 0.001 | Ferritin (µg/L)[$] | 164.0 (42.0, 330.0) | 143.0 (59.0, 410.0) | 0.900 |
| Dyslipidaemia[&] | 45 (40.5) | 15 (34.1) | 0.460 | Chest pain[&] | 17 (15.3) | 8 (18.2) | 0.660 | CRP (mg/L)[$] | 1.4 (0.3, 4.0) | 1.2 (0.4, 2.9) | 0.910 |
| Type II Diabetes[&] | 18 (16.2) | 11 (25.0) | 0.210 | Dysgeusia[&] | 2 (1.8) | 1 (2.3) | 0.850 | **Chest X-ray, day** | 51.4 (27.4) | 56.7 (33.3) | |
| Asthma[&] | 17 (15.3) | 3 (6.8) | 0.150 | Anosmia[&] | 1 (0.9) | 3 (7.0) | 0.034 | Normal X-ray[&] | 71 (64) | 25 (55.6) | 0.130 |
| Smoke habit[&] | 10 (9.0) | 2 (4.5) | 0.630 | GI symptoms[&] | 23 (20.7) | 18 (40.9) | 0.010 | **COVID-19 Complications** | | | |
| Chronic Kidney Disease[&] | 10 (9.0) | 1 (2.3) | 0.140 | **Number of symptoms** | | | | ICU entry[&] | 1 (0.9) | 3 (6.8) | 0.036 |
| Cancer[&] | 8 (7.2) | 2 (4.5) | 0.540 | None[&] | 27 (24.3) | 0 (0.0) | <0.001 | Home Oxygen Therapy[&] | 5 (4.5) | 3 (6.8) | 0.560 |
| COPD[&] | 5 (4.5) | 1 (2.3) | 0.520 | ≤ 3[&] | 63 (56.8) | 24 (54.5) | <0.001 | Readmission[&] | 6 (5.4) | 3 (6.8) | 0.730 |
| Heart failure[&] | 5 (4.5) | 1 (2.3) | 0.520 | ≥ 4[&] | 21 (18.9) | 20 (45.5) | <0.001 | Thromboembolism[&] | 2 (1.8) | 4 (9.1) | 0.034 |

&: number (%)

#: mean (standard deviation)

$: median (interquartile range), BMI (body mass index), COPD (chronic obstructive pulmonary disease), GI (gastrointestinal), ICU (intensive unit care), CRP (C reactive protein).

weeks COVID-19 course or >4 weeks COVID-19 course. In this study, 28.3% (CI 95% 0.21–0.35) of patients suffered symptoms >4 weeks, especially women (63.6%) and younger patients (55.7 years old). Their symptoms lasted longer, median: dyspnoea 46 days, cough 30 days, asthenia 29 days; especially in those who needed oxygen, median: dyspnoea 101 days, asthenia 61 days. Embolism was more frequent in patients who persisted symptoms >4 weeks than in patients who had symptoms ≤ 4 weeks (9.1% vs 1.8%, p value 0.034). Patients who had symptoms >4 weeks were followed-up for a longer time (median of 81 vs 50 days, p value <0.001) and 6.8% (n: 3) still needed oxygen at the end of the study, p value 0.560. Having symptoms >4 weeks was associated with the duration of dyspnoea, cough and asthenia (S2 File).

However, following NICE definition of long COVID, we explored characteristics of cut-off point at week 12 (patients with persisting symptoms after 12 weeks, compared to patients with ≤12 weeks: S3 File. Patients with >12 weeks of evolution presented more frequently medical

history of COPD (23.1% vs 3.5%, *p* value 0.002), and some COVID-19 symptoms like dyspnoea (76.9% vs 26.8%, *p* value <0.001), myalgia (38.5% vs 12.1%, *p* value 0.009), asthenia (53.8% vs 23.9%, *p* value 0.019) and anosmia (15.4% vs 1.4%, *p* value 0.003). No differences were found in complementary exploratory exams, neither chest X-ray nor blood test. But more medical resources were found with significant difference, as ICU admission (15.4% vs 14%, *p* value 0.002), thromboembolism (15.4% vs 2.8%, *p* value 0.025) and primary care follow-up (phone calls mean 17.9 (SD 10.9) vs 11.6 (SD 5.7), *p* value <0.001).

## Discussion

This study described the clinical course of SARS-CoV-2 pneumonia from primary care focusing on the evolution after 4 weeks of clinical course. The main persistent symptoms were cough, dyspnoea, asthenia and gastrointestinal disturbances. The evolution of the whole illness lasted a median of 54.0 days (IQR 42.0, 88.0) until symptoms disappeared. GPs and nurses follow-up consisted of 12.2 mean calls (SD 6.4). Those who developed persisting COVID-19 symptoms after 4 weeks of onset (28.3%) had more complications (ICU admission and thromboembolic event) and required closer monitoring than those with < 4 weeks COVID-19 symptoms. The need of oxygen therapy was the more serious sequelae of the disease, although there was no statistical significance with ≤ 4 weeks COVID-19 symptoms.

There are few data available on patients who persisted symptoms >4 weeks at the moment in primary care. On the acute phase, Tenforde *et al.* interviewed 270 symptomatic outpatients who had a positive test for SARS-CoV-2 [5]. GPs phoned 2–3 weeks after the positive results were received and patients reported cough, fatigue, congestion or dyspnoea after the acute phase. Most studies collected symptoms 2–3 months after discharge. Carfi *et al.* described patients who were followed-up for 60 days and presented fatigue (53.0%), dyspnoea (43.0%) and joint pain (27.0%) as the most frequent symptoms [4]. Zhao *et al.* described symptoms 3 months after discharge in 55 patients: gastrointestinal symptoms (30.9%), headache (18.1%), fatigue (16.3%) and dyspnoea (14.5%) [18]. Arnold *et al.* described the follow-up of 110 patients [19] and collected symptoms 2–3 months after admission, symptoms were more frequent in those with moderate (75.0%) and severe COVID-19 (89.0%), describing mainly breathlessness, fatigue, myalgia and insomnia. Also, pulmonary and cardiac conditions have been described during recovery [19, 20]. Other studies describe fatigue, dyspnoea, cough as the main symptoms persisting after acute phase [21, 22]. Then, severity in COVID-19 acute phase seems to be related with remaining symptoms. Our results are in concordance with those published data where main symptoms observed were cough, asthenia and dyspnoea. However, Huang *et al* [23] described fatigue or muscle weakness as the main symptom (63.0%) followed by dyspnoea (26.0%), sleep difficulties (26.0%) and hair loss (22.0%) in patients who were discharged at median day 186 of follow-up. Differences with our results can be related with data collection as we considered all range of COVID symptoms following WHO document [1].

The workload of acute (patients who had symptoms ≤ 4 weeks) or long (patients who persisted symptoms >4 weeks) course should be considered in day by day practice. Zheng *et at* recommend remote assessment on day 3 with face to face follow-up on days 7, 14 and 30 after discharge [24]. Subsequently, they suggest a remote assessment on months 3 and 6. Greenhalhg *et al.* recommend physical examination, daily pulse oximetry in case of respiratory symptoms, blood test based on patient's symptoms but not a routine test for each patient [10]. They supported the British Thoracic Society guidelines whom recommend a routine follow-up chest radiograph at 12 weeks after discharge [25]. Raghu *et al* hypothesised that patients admitted to hospital without intensive care would have a 12 months course until pulmonary

abnormalities resolved [26]. Crameri *et al.* observed reduced aerobic capacity in young adults 45 days after symptomatic COVID-19 [27]. They also performed a chest X-ray after 2–3 months of the acute phase, they found normal chest X-ray in 86.0% of patients. Mandal *et al* [22] described the proportion of normal chest X-ray as 62% in concordance with our findings. Persisting chest X-ray abnormalities can be related with severity of COVID-19 acute phase. Then, chest-X ray should be considered after 2–3 months of acute COVID in the follow-up.

On the other hand, if we consider laboratory tests performed, Mandal *et al.* [22] described normal blood test at 4–6 weeks after discharge. Only patients that had abnormal biomarkers at discharge still had them elevated in the follow-up (30.1% for d-dimer and 9.5% for C reactive protein). Their laboratory results resembled our own results and supports the idea that laboratory tests normalized after the acute phase.

Another aspect to be considered is the use of oxygen after discharge. Weerahandi *et al.* studied patients after hospital discharge (n: 148) and found that 35.1% of patients (n: 52) were sent home with oxygen after hospital discharge. They reviewed patients with a median of 37 days and 13.5% (n: 20) still had (or needed) oxygen [6]. Our data found less oxygen therapy, at discharge we had 5.2% (n: 8) with oxygen at home and only 2 patients still had oxygen by the end of the follow-up, may be in concordance with a longer follow-up and recovery of pulmonary affection.

Finally, Kingstone *et al.* described the difficulties these patients found in receiving proper care and also the importance of finding a GP who follow-up the whole condition [28]. Then, more research should be conducted in primary care following patients who persisted symptoms >4 weeks to define properly complementary examinations and visits in pandemic time. We would like to address the importance of defining long COVID. We described SARS-CoV-2 pneumonia recovery in weeks; as expected persisting symptoms (long COVID) decreased with follow-up. When comparing differences in long COVID at 12 weeks of onset (S3 File) and 4 weeks (Table 2) we can observe that most cases are recovered before 12 weeks. And those who reminded symptomatic after 12 weeks had persisting dyspnoea, asthenia and anosmia. As clinicians, we think it is important to determine characteristics of those people who persist with symptoms after 4 weeks from the onset to address new strategies for the follow-up in primary care. We consider week 4 as a milestone especially addressing patient´s perspective when symptoms are not solved as quick as they expected, but we need more studies to offer an evidence-based protocol to follow-up patients in primary and secondary care. On the other hand, follow-up in primary care was 54.0 median days (IQR 42.0, 88.0) in our study, equivalent to 7.7 weeks. So, defining long COVID as persisting symptoms until 12 weeks could be a valid cut off point, in concordance with NICE definition [11]. Then, data on long COVID should be presented with different cut-offs until consensual definition is reached.

## Strengths and limitations

The main strength of this study was to describe the acute phase (patients who had symptoms ≤ 4 weeks) and the long course (patients who persisted symptoms >4 weeks) of SARS-CoV-2 pneumonia in primary care from the initial symptoms to the resolution. We explored the cut-off points to describe long COVID and we described the population of those who still had symptoms >4 weeks. However, some limitations should be considered. An underreporting in data from the electronic health records could be found because of the heavy workload during the pandemic. We collected all medical record from primary health and hospital courses to minimize information bias. Possible differences related with medical treatment in hospital stay was considered, but no differences were detected in preliminary analysis. In addition, we have only registered embolism and re-admission as complications when there

could be more conditions associated with COVID-19. We realized that emotional distress was present in many of them but study designed was focused on physical symptoms.

### Implications for clinical practice and future research

This study has described the clinical course of SARS-CoV-2 pneumonia in patient who persisted symptoms >4 weeks and >12 weeks, both milestones could be a practical cut-off for defining long COVID. However, more qualitative and quantitative research need to be considered before deciding a cut-off taking into account the impact on patients' lives. The quality of life, emotional distress, economic impact and patient´s narrative should be collected in new studies as the need of oxygen at long-term. Stablishing protocols in primary care considering this study results will optimize resources and will improve patient's attention. Routine laboratory tests may not be routinely necessary. And chest X-ray could be considered in those patients with long COVID but not in those recovered in ≤ 4 weeks.

### Conclusion

Most patients with SARS-CoV-2 pneumonia were asymptomatic in ≤4 weeks but persisting COVID-19 symptoms over 4 weeks is a common condition. Long COVID could be defined as those patients who suffered persisting COVID-19 symptoms between 4 and 12 weeks after onset. The role of practices in monitoring COVID-19 patients seems crucial during the pandemic, but it involves a considerable workload to the daily routine of the practice. Future research should aim to establish a clear follow-up to help GPs and nurses to care for these patients.

### Supporting information

**S1 File. Clinical characteristics of patients with SARS-CoV-2 pneumonia follow-up in primary care during 6 month, stratified by age groups (2020).**
(DOCX)

**S2 File. Logistic regression analyses of the influence of suffering long COVID (>4 weeks symptoms) on patient´s characteristics and symptoms.**
(DOCX)

**S3 File. Differences between <12 weeks and ≥12 weeks of clinical course of SARS-CoV-2 pneumonia patients.**
(DOCX)

### Acknowledgments

Ana Herrero and Ana Herrera who were part of the team at the beginning of this research. E. García and S. Collen provided professional writing services. We would like to thank E. Polentinos and M. Menéndez for guidance in the revision of the manuscript. We would like to thank all the healthcare professionals who worked at A&E Department Hospital Infanta Leonor and Federica Montseny Health Centre. This work could not be done without their effort and dedication.

### Author Contributions

**Conceptualization:** Sara Ares-Blanco, Marta Pérez Álvarez, Ileana Gefaell Larrondo, Cristina Muñoz, Vanesa Aguilar Ruiz, Marina Guisado-Clavero.

**Data curation:** Marta Pérez Álvarez, Ileana Gefaell Larrondo, Cristina Muñoz, Marta Castelo Jurado.

**Formal analysis:** Sara Ares-Blanco, Marina Guisado-Clavero.

**Funding acquisition:** Sara Ares-Blanco.

**Investigation:** Marta Pérez Álvarez, Ileana Gefaell Larrondo, Cristina Muñoz, Vanesa Aguilar Ruiz, Marta Castelo Jurado, Marina Guisado-Clavero.

**Methodology:** Sara Ares-Blanco, Marina Guisado-Clavero.

**Project administration:** Sara Ares-Blanco, Marina Guisado-Clavero.

**Supervision:** Sara Ares-Blanco, Marina Guisado-Clavero.

**Validation:** Sara Ares-Blanco, Marta Castelo Jurado, Marina Guisado-Clavero.

**Visualization:** Marta Pérez Álvarez, Vanesa Aguilar Ruiz.

**Writing – original draft:** Sara Ares-Blanco, Marta Pérez Álvarez, Marta Castelo Jurado, Marina Guisado-Clavero.

**Writing – review & editing:** Sara Ares-Blanco, Ileana Gefaell Larrondo, Cristina Muñoz, Vanesa Aguilar Ruiz, Marina Guisado-Clavero.

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
