## [Decision Letter · Decision Letter 0]

9 Jun 2021

PONE-D-21-09441

SARS-CoV-2 pneumonia follow-up and long COVID in Primary care: a retrospective observational study in Madrid City

PLOS ONE

Dear Dr. Guisado-Clavero,

Thank you for submitting your manuscript to PLOS ONE. After careful consideration, we feel that it has merit but does not fully meet PLOS ONE’s publication criteria as it currently stands. Therefore, we invite you to submit a revised version of the manuscript that addresses the points raised during the review process.

Please review comments made by reviewers and provide point by point response in your revised manuscript.

We look forward to receiving your revised manuscript.

Kind regards,

Muhammad Adrish, MD, MBA, FCCP, FCCM

Academic Editor

PLOS ONE

Journal Requirements:

2. In your ethics statement in the Methods section and in the online submission form, please provide additional information about the data used in your retrospective study. Specifically, please ensure that you have discussed whether all data were fully anonymized before you accessed them and/or whether the IRB or ethics committee waived the requirement for informed consent. If patients provided informed written consent to have data from their medical records used in research, please include this information.

Reviewers' comments:

Reviewer's Responses to Questions

**Comments to the Author**

1. Is the manuscript technically sound, and do the data support the conclusions?

Reviewer #1: Yes

Reviewer #2: Yes

2. Has the statistical analysis been performed appropriately and rigorously? 

Reviewer #1: Yes

Reviewer #2: Yes

3. Have the authors made all data underlying the findings in their manuscript fully available?

Reviewer #1: No

Reviewer #2: Yes

4. Is the manuscript presented in an intelligible fashion and written in standard English?

Reviewer #1: Yes

Reviewer #2: Yes

5. Review Comments to the Author

Reviewer #1: The authors present a thorough discussion of the duration of COVID19 symptoms following acute infection.

While this is a useful contribution, the terminology uses is imprecise and unclear, particularly given the varying and non-standardized definitions of "long-COVID". Additionally, as the pathophysiology of this process is not understood, a purely descriptive report in which the average duration of different clinical symptoms are reported, without the addition of syndromic terminology.

Reviewer #2: First I would like to thank the authors on their efforts in writing this manuscript about describing COVID-19 pneumonia and identify the cut offs to define long covid in primary care patients.

Please see my comments.

Intro-

Overall good introduction to the topic and literature review.

The research aimed to determine patient and disease characteristics associated with SARS- CoV-2 pneumonia since the pneumonia diagnosis until recovery in a study population aged 18 to 91 years during the follow-up in primary care. The secondary outcome was to describe patients who still had symptoms 4 weeks after the diagnosis of SARS-CoV-2 pneumonia.

I would recommend the authors to revise the aim to be clearer. From the abstract, results and findings, it seems its more on describing characteristics as well as identifying/evaluating the cut-offs for long COVID in primary care? Please clarify and revise the statement in intro to be more consistent and clearer as a purpose/aim for the study.

Methods:

Line 102= “patient with lung affection by infiltration…” – do you mean infection? Please clarify.

It was a little bit difficult to follow methods although it was very well detailed and described. However, from a flow perspective it was a little bit difficult. In the selection of study subjects, authors mentioned that about 1023 patients were seen in the PHCC clinic, and the sample was limited to 10th of march to 7th of April 2020. Later in line 103- it says that data was collected 6 months after the diagnosis. Just to clarity, was the sample of 155 (that was included in the study, were seen during that one month period 3/10 to 4/7 2020? Please clarify as it is not clear.

Results-

I am not sure if data of the excluded patients should be included since the study has a pre-defined inclusion criteria in which only 155 patients qualified. Thus, data in supplementary file might not be needed from an exclusion criteria perspective.

I would recommend the authors to move the long COVID definition and describe all of its aspects from definition and cut off points in the methodology section (excluding the reporting data). Then data and results can be presented in the results section. The purpose of the results section to report the data without interpretation nor definitions.

Figure 1 – the figure is nice as it captures the study time line. However, modification is strongly recommended. For example, baseline time for blood test is not identified. In addition, the presented time line is not clear. Does it describe 0 from diagnosis (march and April) and then all the way to 12 weeks? If so, I would recommend adding a horizontal line that shows mean with IQR on top of the hospital LOS median that way its more clear.

Looking at the two symptoms figure, please describe the values (i.e. Fever, is that %? Of so,

please add the total and % in the figure legend or inside the figure it self.

Table 3- regarding phone calls follow up , does that reflect number of patients who had /received phone calls? Or number of times the physician/nurse called? Please clarify.

6. PLOS authors have the option to publish the peer review history of their article (what does this mean?). If published, this will include your full peer review and any attached files.

Reviewer #1: **Yes: **John Cafardi

Reviewer #2: No

---

## [Author Response · Author response to Decision Letter 0]

1 Jul 2021

ANSWERS TO THE REVIEWRS:

**Reviewer 1**: The authors present a thorough discussion of the duration of COVID19 symptoms following acute infection.

While this is a useful contribution, the terminology uses is imprecise and unclear, particularly given the varying and non-standardized definitions of "long-COVID". Additionally, as the pathophysiology of this process is not understood, a purely descriptive report in which the average duration of different clinical symptoms are reported, without the addition of syndromic terminology.

ANSWER: We are grateful for the commentary and we agree as there is not a unique definition of long COVID. We decided to change the terminology to ≤4 weeks and >4 weeks to avoid confusion. Changes are located throughout the all manuscript (abstract in lines 48-52, results section in lines 170-175; discussion section in lines 190-197 and line 215, 241, 256-257).

**Reviewer 2**: First I would like to thank the authors on their efforts in writing this manuscript about describing COVID-19 pneumonia and identify the cut offs to define long covid in primary care patients.

- Intro: Overall good introduction to the topic and literature review. The research aimed to determine patient and disease characteristics associated with SARS- CoV-2 pneumonia since the pneumonia diagnosis until recovery in a study population aged 18 to 91 years during the follow-up in primary care. The secondary outcome was to describe patients who still had symptoms 4 weeks after the diagnosis of SARS-CoV-2 pneumonia.

I would recommend the authors to revise the aim to be clearer. From the abstract, results and findings, it seems its more on describing characteristics as well as identifying/evaluating the cut-offs for long COVID in primary care? Please clarify and revise the statement in intro to be more consistent and clearer as a purpose/aim for the study.

ANSWER: We agree about your commentary. We have cleared up the aims to give a clearer objective.

We have added (introduction section line 87-91): “The research aimed to determine patient and disease characteristics associated with SARS-CoV-2 pneumonia since the pneumonia diagnosis until recovery in a study population aged 18 to 91 years during the follow-up in primary care especially focusing on patients who still had symptoms 4 weeks after SARS-CoV-2 pneumonia diagnosis. The secondary outcome was to describe the different cut-off points of long COVID definition in week 4 and in week 12. “

- Methods:

Line 102= “patient with lung affection by infiltration…” – do you mean infection? Please clarify.

It was a little bit difficult to follow methods although it was very well detailed and described. However, from a flow perspective it was a little bit difficult. In the selection of study subjects, authors mentioned that about 1023 patients were seen in the PHCC clinic, and the sample was limited to 10th of march to 7th of April 2020. Later in line 103- it says that data was collected 6 months after the diagnosis. Just to clarity, was the sample of 155 (that was included in the study, were seen during that one month period 3/10 to 4/7 2020? Please clarify as it is not clear.

ANSWER: Thank you for your observations. For a better comprehension, we have proceed to modify the following points

- Line 104 (previous line 102) has been changed to “Patients with lung affection by SARS-CoV-2 infection” to clarify the concept. 

- We have added a flow chart (Figure 1) to simplify the flow of patients in this study. In the manuscript, methods section line 93-96, has been added the following explanation “The PHCC provided care to 21814 population, from whom 1,023 patients were diagnose from COVID-19 (confirmed by RCP or symptoms compatible with COVID-19), from the 10th of March to the 7th of April of 2020. Pneumonia was present in 251 and 155 were diagnosed and followed-up by Federica Montseny PHCC during 6 months (Figure 1).” We have also added it in the eligibility criteria (methods section, line 107): “Eligibility criteria were considered in patients ≥ 18 years, followed-up by the primary care practice during the next 6 months after SARS-CoV-2 pneumonia diagnosis, with a suspicious of SARS-CoV-2 pneumonia and a radiologist report which confirmed the SARS-CoV-2 pneumonia”. 

- Results: I am not sure if data of the excluded patients should be included since the study has a pre-defined inclusion criteria in which only 155 patients qualified. Thus, data in supplementary file might not be needed from an exclusion criteria perspective.

ANSWER: We have removed Supplementary file 1 as the reviewer suggested, we wanted to show the maximum transparency, which was the reason for including it. If the editor thinks the material should be included, we are pleased to include it again. All Supplementary files order has been reorganized thought the manuscript according to the exposed change. 

I would recommend the authors to move the long COVID definition and describe all of its aspects from definition and cut off points in the methodology section (excluding the reporting data). Then data and results can be presented in the results section. The purpose of the results section to report the data without interpretation nor definitions.

ANSWER: We agree with the commentary and we are grateful for the opinion shared. We have described the methodology and results as “patients who had symptoms ≤ 4 weeks” or patients “who persisted symptoms >4 weeks” to avoid misinterpretation of the data (abstract in lines 48-52, results section in lines 170-175; discussion section in lines 190-197 and line 215, 241, 256-257).

- Figure 1: The figure is nice as it captures the study time line. However, modification is strongly recommended. For example, baseline time for blood test is not identified. In addition, the presented time line is not clear. Does it describe 0 from diagnosis (March and April) and then all the way to 12 weeks? If so, I would recommend adding a horizontal line that shows mean with IQR on top of the hospital LOS median that way its more clear. 

Looking at the two symptoms figure, please describe the values (i.e. Fever, is that %? Of so, please add the total and % in the figure legend or inside the figure it self.

ANSWER: Thank you for your recommendations. A new version of this figure has been uploaded for a better understanding of this primary-care follow-up sample considering your observations. After previous changes, this figure has been labelled as Figure 2. 

- Table 3: regarding phone calls follow up, does that reflect number of patients who had /received phone calls? Or number of times the physician/nurse called? Please clarify.

ANSWER: The phone calls reflect the number of times the physician or nurse made to follow-up the patient. We have changed in Table 2 the title to: “Number of phone calls by GP or nurse during the follow-up”.

---

## [Decision Letter · Decision Letter 1]

6 Sep 2021

SARS-CoV-2 pneumonia follow-up and long COVID in Primary care: a retrospective observational study in Madrid City

PONE-D-21-09441R1

Dear Dr. Guisado-Clavero,

We’re pleased to inform you that your manuscript has been judged scientifically suitable for publication and will be formally accepted for publication once it meets all outstanding technical requirements.

Kind regards,

Muhammad Adrish, MD, MBA, FCCP, FCCM

Academic Editor

PLOS ONE

Additional Editor Comments (optional):

All comments have been addressed

Reviewers' comments:

Reviewer's Responses to Questions

**Comments to the Author**

1. If the authors have adequately addressed your comments raised in a previous round of review and you feel that this manuscript is now acceptable for publication, you may indicate that here to bypass the “Comments to the Author” section, enter your conflict of interest statement in the “Confidential to Editor” section, and submit your "Accept" recommendation.

Reviewer #1: All comments have been addressed

Reviewer #2: All comments have been addressed

2. Is the manuscript technically sound, and do the data support the conclusions?

Reviewer #1: Yes

Reviewer #2: Yes

3. Has the statistical analysis been performed appropriately and rigorously? 

Reviewer #1: I Don't Know

Reviewer #2: Yes

4. Have the authors made all data underlying the findings in their manuscript fully available?

Reviewer #1: No

Reviewer #2: Yes

5. Is the manuscript presented in an intelligible fashion and written in standard English?

Reviewer #1: Yes

Reviewer #2: Yes

6. Review Comments to the Author

Reviewer #1: (No Response)

Reviewer #2: The authors have address the reviewers comments. The modified manuscript with the new figure as well as adding the logistic regression is good to have for clarity of the manuscript.

7. PLOS authors have the option to publish the peer review history of their article (what does this mean?). If published, this will include your full peer review and any attached files.

Reviewer #1: No

Reviewer #2: No

---

## [Editor Report · Acceptance letter]

14 Sep 2021

PONE-D-21-09441R1 

SARS-CoV-2 pneumonia follow-up and long COVID in Primary care: a retrospective observational study in Madrid City 

Dear Dr. Guisado-Clavero:

I'm pleased to inform you that your manuscript has been deemed suitable for publication in PLOS ONE. Congratulations! Your manuscript is now with our production department. 

Kind regards, 

on behalf of

Dr. Muhammad Adrish 

Academic Editor

PLOS ONE